# Multidimensional Effects of Manual Therapy Combined with Pain Neuroscience-Based Sensorimotor Retraining in a Patient with Chronic Neck Pain: A Case Study Using fNIRS

**DOI:** 10.3390/healthcare13141734

**Published:** 2025-07-18

**Authors:** Song-ui Bae, Ju-hyeon Jung, Dong-chul Moon

**Affiliations:** 1Department of Physical Therapy, Graduate School, Dong-Eui University, Busan 47340, Republic of Korea; song2732@naver.com; 2Department of Physical Therapy, College of Nursing, Healthcare Sciences and Human Ecology, Dong-Eui University, Busan 47340, Republic of Korea; 3Department of Physical Therapy, Gimhae College, Gimhae-si 50811, Republic of Korea; ptmdc@gh.ac.kr

**Keywords:** chronic pain, pain management, near-infrared spectroscopy, manual therapy

## Abstract

Chronic neck pain is a multifactorial condition involving physical, psychological, and neurological dimensions. This case report describes the clinical course of a 25-year-old female with chronic neck pain and recurrent headaches who underwent a 6-week integrative intervention consisting of manual therapy and pain neuroscience-based sensorimotor retraining, administered three times per week. Outcome measures included the Headache Impact Test-6 (HIT-6), Neck Pain and Disability Scale (NPDS), Pain Catastrophizing Scale (PCS), Fear-Avoidance Beliefs Questionnaire (FABQ), pressure pain threshold (PPT), cervical range of motion (CROM), and functional near-infrared spectroscopy (fNIRS) to assess brain activity. Following the intervention, the patient demonstrated marked reductions in pain and psychological distress: HIT-6 decreased from 63 to 24 (61.9%), NPDS from 31 to 4 (87.1%), FABQ from 24 to 0 (100%), and PCS from 19 to 2 (89.5%). Improvements in PPT and CROM were also observed. fNIRS revealed decreased dorsolateral prefrontal cortex (DLPFC) activation during pain stimulation and movement tasks, suggesting a possible reduction in central sensitization burden. These findings illustrate that an integrative approach targeting biopsychosocial pain mechanisms may be beneficial in managing chronic neck pain, improving function, and modulating cortical responses. This report provides preliminary evidence in support of the clinical relevance of combining manual therapy with neurocognitive retraining in similar patients.

## 1. Introduction

Chronic neck pain (CNP) is a prevalent health issue affecting over 30% of adults globally [1]. The condition extends beyond mere physical discomfort, as it is often accompanied by diminished work productivity and psychological challenges, including sleep disturbance, anxiety, and depression, severely impairing patients’ functional independence and quality of life [2]. These issues impose a substantial burden on healthcare systems and societal productivity, underscoring the necessity for effective, sustainable management and treatment strategies [3].

CNP is a multifactorial condition influenced by a complex interplay of biological, psychological, and social factors, with central sensitization triggered by repeated pain stimuli and psychological stress [4]. This sensitization represents a hypersensitive response of the central nervous system, leading to excessive pain even in the absence of actual stimuli, and serves as a major barrier to effective treatment [5]. Biological interventions such as manual therapy (MT) are known to be effective for restoring joint and soft tissue function and providing short-term pain relief [6]. However, in patients with chronic musculoskeletal pain, the effect of MT on central sensitization is limited, and its efficacy and level of evidence for long-term pain regulation are reported to be low [7,8].

Pain neuroscience-based sensorimotor retraining helps patients understand that pain is not solely the result of tissue damage but can also arise from central sensitization and nervous system hypersensitivity [9]. Through this process, patients begin to reinterpret pain not as a direct ‘danger signal’ but as a modifiable response of the nervous system, leading to changes in their previous pain beliefs and promoting cognitive reappraisal [10]. The approach focuses on educating patients that pain is not solely a consequence of tissue injury, but a complex process involving CNS sensitization [11]. Consequently, patients can dispel erroneous beliefs and fears regarding pain and actively participate in pain management and rehabilitation [12]. Previous studies have reported that pain neuroscience-based sensorimotor retraining effectively reduces patients’ pain sensitivity and alleviates psychological factors such as pain catastrophizing, thus improving functional independence and quality of life [9]. It provides a more integrative, effective treatment approach for chronic pain management by addressing the mechanisms of central sensitization [13]. Specifically, as a treatment capable of integratively addressing the multi-dimensional factors contributing to pain, pain neuroscience-based sensorimotor retraining demonstrates potential in overcoming the limitations of existing treatment modalities [14].

Meanwhile, recent advancements in neuroimaging technology are creating new avenues for pain research [15]. Functional near-infrared spectroscopy (fNIRS) is an innovative technique for indirectly measuring brain activity by measuring light absorption and calculating changes in oxygenated hemoglobin (HbO) and deoxygenated hemoglobin (HbR) concentrations [16]. It facilitates real-time non-invasive measurements of changes in HbO in the brain, enabling quantitative assessment of activity changes in key brain areas related to pain sensitivity [17]. The prefrontal cortex (PFC) is particularly significant for pain sensitization and psychological regulation, and the dorsolateral prefrontal cortex (DLPFC) is involved in attentional shift, pain inhibition, and emotional regulation [17,18]. Previous research has identified abnormal activity patterns in the DLPFC in cases of CNP, indicating a neurological mechanism associated with pain sensitization [19,20]. DLPFC dysfunction compromises the inhibitory effects of the descending pain modulatory system, leading to heightened sensitivity to pain stimuli and playing a key role in pain chronicization and persistence [21].

Consequently, this study aimed to evaluate the effects of an integrative treatment approach combining MT with pain neuroscience-based sensorimotor retraining and analyze changes in DLPFC activity using fNIRS. We hypothesized that this integrative approach could ameliorate pain mechanisms, including central sensitization, and enhance the quality of life in patients with CNP.

## 2. Case Presentation

The patient, a 25-year-old woman, was diagnosed with panic disorder in the 9th grade (at age 15) and had subsequently experienced chronic insomnia. Initially, her symptoms were managed with anxiolytic medication, and after the age of 20, her treatment was switched to hypnotic medication, which has effectively controlled her insomnia to date. Due to clinical suspicion of autonomic hyperactivity, she was evaluated by a neurologist. Through electroencephalography (EEG) and neurological examination, structural abnormalities such as disc herniation or radiculopathy were ruled out. EEG results revealed signs of autonomic hyperactivity, suggesting that the patient’s symptoms were more likely related to central sensitization and dysfunction in pain modulation rather than structural damage.

Over the past 2 years, the patient had experienced persistent pain in the neck and shoulders, accompanied by occipital paresthesia. The pain worsened with cervical extension, left lateral flexion, and prolonged use of the left arm. These symptoms were frequently accompanied by headaches radiating from the occipital to the temporal regions, especially after prolonged sitting or intensive tasks, significantly interfering with her daily functioning. In the last six months, the pain progressively worsened, spreading to the scapular region, and was further aggravated by specific postures or physical activity (Table 1).

The study was conducted according to the Declaration of Helsinki. Prior to participation, the patient provided written informed consent for the research objectives, as well as for the use of photographs and experimental data. Ethical approval was obtained from the Dong-Eui University Institutional Review Board (approval number DIRB-202408-HR-E-40) before the commencement of the study.

### 2.1. Procedures

The intervention for CNP was conducted over a total of 8 weeks, with three sessions per week. It consisted of manual therapy, contemporary pain-science-based education, graded pre-movement training, and graded movement and loading training. Following the intervention period, a 2-week self-exercise program was implemented.

To assess the effect of the treatment on the CNP, the six-item Headache Impact Test (HIT-6), Neck Pain and Disability Scale (NPDS), Fear-Avoidance Beliefs Questionnaire (FABQ), and Pain Catastrophizing Scale (PCS) were administered before and after the intervention, and the pressure pain threshold (PPT), cervical range of motion (CROM), and fNIRS were sequentially measured as well. These tests and measurements were repeated weekly throughout the 6-week intervention period. Following the intervention, a 2-week self-exercise program was implemented, and the tests were repeated to determine whether the effects of the intervention persisted. Specifically, the patient wore the fNIRS apparatus to analyze neural activity during PPT and range of motion (ROM) measurement, enabling combined assessment of physiological and neurological adaptation pre- and post-treatment. The HIT-6 and NPDS were employed to evaluate functional impairment due to headache and cervical pain, while FABQ and PCS were used to analyze changes in psychological factors associated with pain (Figure 1).

### 2.2. Assessments

The PPT was measured using a digital pressure algometer (Wagner Instruments, Greenwich, CT, USA), an instrument used to quantify pain sensitivity. The patient was instructed to respond immediately when the pressure was perceived as pain, and this point was recorded on a numerical scale. After taking three measurements each from both the upper trapezius (UT) muscles, the mean score for each side was calculated and used in the analysis [22].

The CROM was analyzed using a digital inclinometer (J TECH Medical, Midvale, UT, USA) to measure the ranges of cervical flexion, extension, side-bending (R/L), and rotation (R/L). The ROM serves as a key index for evaluating cervical function and restricted movement. The extent of functional recovery was analyzed by comparing changes in ROM pre- and post-intervention [23].

fNIRS is a non-invasive neuroimaging technique used to analyze changes in brain activity. In this study, an fNIRS device (OBELAB Inc., Seoul, Korea) was used to measure real-time changes in blood flow and oxygenation in the PFC and DLPFC. fNIRS precisely recorded brain activity during pain stimulation and exercise performance to analyze CNS changes related to pain processing and motor control [24]. Measurements were obtained using a wearable fNIRS device, which was secured to the head using an elastic strap and plastic cap. The central marker of the device was aligned over the nasion, according to the international 10–20 system, and the cap was adjusted to position the lower margin immediately above the participant’s eyebrows. The system employs 24 light sources (maximum power < 1 mW) at two different wavelengths (780 nm and 850 nm) and 32 photo-detectors to measure PFC signals from the PFC. Data were collected from 48 channels spaced 3 cm apart. The detected signals were preprocessed by applying low-pass (DCT 0.1 Hz) and high-pass filters (DCT 0.005 Hz) to minimize noise due to movement and environmental light interference [24]. Prior to fNIRS measurements, multiple calibration procedures were performed to ensure the accuracy and reliability of the signal and the equipment. First, gain calibration was conducted to optimize the sensitivity of each light source and detector. This is a crucial step for equalizing signal quality across channels and improving measurement precision. Additionally, motion calibration was carried out using a gyroscope to correct for signal distortions that may occur due to movement during measurement. Furthermore, the collected light signals underwent digital signal processing, converting them into a format suitable for measuring changes in oxygenated and deoxygenated hemoglobin (HbO and HbR). These calibration procedures are designed to account for individual differences such as head shape and skin tone, and they are automatically executed by simply clicking the “Calibration” button within the software.

For each measurement, the signal was recorded for 30 s at rest (baseline), and then the hemodynamic changes were recorded continuously during ROM and PPT testing. For ROM measurement, activity was assessed during extension, side-bending, and rotation, and for the PPT test, changes in neural activity were analyzed during pressure application. Specifically, for the fNIRS measurements taken during PPT, using the pre-intervention PPT results as a reference, changes in brain activity were measured weekly while stimulating the tissue at a pressure equal to the initial PPT values (e.g., right UT: 8.25 Kgf/cm^2^).

The collected fNIRS data were processed and analyzed using the NIRSIT Quest (v1.1.3) software provided by OBELAB Inc. After defining task intervals in the raw data, signal processing was performed to remove spikes due to movement, and the signal was passed through low-pass and high-pass filters and converted to optical density. The modified Beer–Lambert Law was then applied to calculate changes in HbO and HbR (units: mMol). For data analysis, the “Block Average” function was utilized to visualize and quantify mean activity during the performance of each task. Based on the results, changes in the hemodynamic response in the PFC were qualitatively and quantitatively analyzed according to the task type.

The HIT-6 questionnaire was used to analyze the effect of headaches on daily living and quality of life. The HIT-6 measures the frequency and intensity of headaches and has been used to assess the impact of headaches on the patient’s functional activities and quality of life [25].

The NPDS was used to assess neck pain intensity and pain-related disability. The NPDS is capable of multi-dimensional analysis of the extent of functional difficulties and overall disability in daily living caused by neck pain [26].

The FABQ was used to assess fear and avoidance beliefs pertaining to pain and measure the extent to which the patient exhibited avoidance behaviors due to pain. The PCS is a questionnaire that analyzes pain catastrophizing by assessing negative thought patterns related to pain experience [27].

## 3. Interventions

The intervention in this study comprised MT and pain neuroscience-based sensorimotor retraining, provided in 60 min sessions, three times per week, for 6 weeks. All interventions were administered by a licensed physical therapist with over 10 years of clinical experience in musculoskeletal rehabilitation. Upon completion of the 6-week program, the patient was provided with guidelines for self-exercise to be performed independently (Figure 1).

MT was conducted to alleviate myofascial tension and improve joint mobility in the shoulder, reduce pain, and promote functional recovery. Mobilization was applied to enhance cervical joint mobility, and myofascial release was performed to relieve soft tissue tension and promote relaxation [28]. The intervention focused on reducing pain and muscle tension, using trigger point therapy to relieve myofascial tender points, and neurodynamic techniques were used to relieve neural tissue tension and improve neural mobility [29].

Pain neuroscience-based sensorimotor retraining was performed as a multi-dimensional intervention involving pain neuroscience education, graded pre-movement training, graded movement, loading training, and self-exercise [30]. Pain neuroscience education was conducted over a six-week period, focusing on improving the patient’s understanding of various factors influencing pain, including structural, psychosocial, and neurophysiological mechanisms. This was designed to alleviate the patient’s negative beliefs and fear of pain and to encourage active participation in treatment (Figure 2A).

Graded pre-movement training included sensory precision training and mental rehearsal. Sensory precision training focused on improving localization (the ability to perceive the location of stimuli), discrimination (the ability to distinguish small differences between stimuli), and graphesthesia (the ability to recognize letters or figures drawn on the skin) (Figure 2B). Mental rehearsal training uses left–right recognition and motor imagery and aids in improving motor planning and execution ability at the implicit and explicit levels (Figure 2C).

Graded movement and loading training comprised three stages. In the first stage, initial training was performed within the pain-free range, During the second stage, the load and ROM were progressively increased to promote functional recovery, and visual biofeedback training using a laser pointer was incorporated to assist with movement control [31]. In the third stage, the goal was to achieve functional application, encompassing complex movements and real-life activities. The self-exercise program incorporated elements of sensorimotor retraining and graded loading training, aiming to enable the patient to maintain the therapeutic effects in daily living (Figure 2D,E) [14].

## 4. Results

Following the 6-week intervention, patient-reported outcomes—including the HIT-6, NPDS, FABQ, and PCS—showed reductions, and these improvements were maintained or further enhanced at the 2-week follow-up (Table 2, Figure 3). The pressure pain threshold (PPT) increased in both the left and right upper trapezius muscles, with a particularly notable increase on the left side compared to baseline. This improvement was sustained or further enhanced at follow-up (Table 3, Figure 3).

Cervical range of motion (ROM) increased during extension and left side-bending movements, and the gains were preserved at both the 6-week and follow-up time points (Table 3, Figure 3).

fNIRS analysis revealed a general decrease in oxygenation levels in the left and right PFC and DLPFC during extension, side-bending, and pressure stimulation of the upper trapezius (Table 4 and Table 5, Figure 4).

## 5. Discussion

In this case study, we aimed to conduct a multifaceted investigation of the effects of combining manual therapy (MT) with pain neuroscience-based sensorimotor retraining in a patient with chronic neck pain (CNP). The intervention led to notable improvements across multiple domains: reductions in headache severity and neck pain-related disability (as assessed by HIT-6 and NPDS), enhanced psychological outcomes (measured by FABQ and PCS), and increased cervical range of motion. Furthermore, pressure pain thresholds (PPTs) improved, indicating a decrease in mechanosensitivity of the cervical muscles. Most notably, functional near-infrared spectroscopy (fNIRS) revealed decreased oxygenation levels in the dorsolateral prefrontal cortex (DLPFC) and prefrontal cortex (PFC), suggesting a reduction in central sensitization. These findings suggest that the combined intervention effectively modulates pain, function, psychological factors, and cortical activity, providing evidence that integrating MT and pain neuroscience-based retraining may serve as a multidimensional therapeutic approach for managing chronic neck pain.

### 5.1. Headache and Neck Pain-Related Disability

The HIT-6 and NPDS results revealed that functional limitations due to headache and neck pain decreased when MT and pain neuroscience-based sensorimotor retraining were provided concurrently. This indicates that MT may be effective in enhancing cervical mobility and soft tissue relaxation, while pain neuroscience-based sensorimotor retraining might improve the understanding of pain mechanisms and support cognitive restructuring, potentially boosting treatment adherence. In previous studies, MT has been reported to increase myofascial release and joint mobility, reducing chronic cervical pain and headache intensity, while pain neuroscience education has been reported to induce positive changes in pain beliefs and promote functional recovery [32,33]. In particular, the graded pre-movement training (localization, discrimination, and graphesthesia) used in this study encourages patients to perceive and perform movements precisely [31]. Localization training improved body position sense, and discrimination training allowed the patient to more precisely distinguish tactile information, while graphesthesia training provided accurate sensory feedback. This approach improved sensory awareness before exercise and appears to have contributed to pain reduction by increasing movement accuracy.

### 5.2. Changes in Psychological Factors

The FABQ and PCS results showed a decrease in both fear-avoidance belief and pain catastrophizing post-intervention, meaning the patient exhibited a decrease in the tendency to avoid specific activities due to fear of pain and an exaggerated pain perception [34,35]. In previous studies, MT alone had limited effect in improving the FABQ and PCS, suggesting that MT is effective in improving psychological mechanisms but not the psychological factors [36,37]. Meanwhile, MT is likely to be more effective at changing pain beliefs and perceptions when accompanied by sensorimotor training. In particular, pain neuroscience education and therapeutic exercise have been reported to have enhanced effects when applied together [15,38]. This finding suggests that improving the understanding of pain mechanisms could be effective in regulating psychological factors alongside physical retraining. In this study, pain neuroscience-based sensorimotor retraining, mental rehearsal, and imagined movement training were provided in parallel, focusing on improving the patient’s understanding of pain mechanisms and promoting cognitive restructuring. In particular, left–right recognition and implicit and explicit imagery training were applied before exercise to adjust the patient’s negative expectations of pain before performing a said movement. As this process was repeated, negative perceptions of pain and movement gradually decreased, resulting in alleviation of the patient’s fear-avoidance belief and promoting psychological recovery.

### 5.3. PPT and Cervical ROM

In this study, improvements in the PPT and cervical ROM were observed. In previous studies, MT was reported to alleviate soft tissue tension and increase joint mobility, resulting in temporary improvements in ROM [39]. With regard to PPT improvements, myofascial release and a neurodynamic approach have been demonstrated to be effective in reducing pain sensitivity [40,41]. However, the effects of MT have mostly been limited to short-term pain relief and joint mobility, with a trend for the effects to diminish over time [42,43].

Conversely, in the present study, PPT and ROM showed persistent improvements lasting over 6 weeks, indicating that graded movement and loading training play important roles. Because the combination of graded movement and loading training is an approach that extends beyond simply improving mobility and improves neuromuscular control and promotes sensorimotor integration through progressive loading of the tissue, it can be expected to produce a more sustained effect than passive treatment [30]. In this study, in the initial stage of graded movement and loading training, low-intensity movement was performed in the pain-free range to reduce neural hypersensitivity, and then, the load was gradually increased to encourage mechanical adaptation of the muscle and soft tissue. In the later stage, multidirectional movement patterns and complex functional tasks were included to promote neuromuscular adaptation.

The increase in PPT observed using this systematic approach is likely related to the process of graded loading for reducing pain sensitivity. Previous studies have also shown that resistance exercise increases local tissue blood flow and eliminates inflammatory mediators, alleviating peripheral sensitization [44]. Repeated load application is known to promote endogenous analgesia by activating descending pain modulation, and PPT appears to have improved in this study via a similar mechanism [45].

Meanwhile, the increase in ROM can be interpreted as being related to neuromuscular control and mechanical adaptation of the tissue. Repeated loading is thought to have promoted changes in the elastic properties of tissue and improved joint mobility, and as neuromuscular control improved, more precise movements became possible [46]. Specifically, in the later stages of the graded movement and loading training used in this study, in addition to gradual load progression, multidirectional movement patterns and complex functional tasks were included to promote neuromuscular adaptation. This process is likely to have affected the improvement in ROM. In summary, the fact that PPT and ROM both improved in this study indicates that graded movement and loading training reduced tissue sensitivity, improved neuromuscular control, and restored functional movement.

### 5.4. Neurological Changes (fNIRS: During Extension, Side Bending, and PPT)

This study observed changes in oxygenation within the prefrontal cortex (PFC) using functional near-infrared spectroscopy (fNIRS). As the intervention progressed, a gradual decrease in cortical oxygenation and a concurrent increase in pressure pain threshold (PPT) were noted. These findings suggest reduced pain sensitivity to identical stimuli and a reduction in the functional load on cortical areas involved in pain modulation [47,48]. In particular, during extension and left lateral flexion tasks, oxygenation levels in the dorsolateral prefrontal cortex (DLPFC) decreased, coinciding with reductions in Fear-Avoidance Beliefs Questionnaire (FABQ) scores [49,50,51]. Given that higher FABQ scores are generally associated with DLPFC hyperactivation, the observed decrease in oxygenation, in this case, may reflect attenuation of central sensitization and recalibration of pain perception.

Transient increases in PFC and DLPFC oxygenation observed at specific time points could be interpreted as adaptive responses to the uncertainty of new motor learning and pain prediction [52]. In the early stages of treatment, activation of the PFC tends to be widespread; however, with increased task familiarity and emotional stabilization, this pattern may shift toward more efficient neural activation characterized by selective engagement of relevant cortical regions [53,54].

The selective reactivation observed during lateral flexion in the later phase of treatment appears to reflect strategic pain modulation rather than mere inhibition, aligning with recovery patterns reported in cognitive behavioral therapy and motor rehabilitation [55,56]. The DLPFC, in particular, is a key area involved in top-down pain inhibition, attentional control, and emotional regulation [48]. Its reactivation in this study may suggest that patients were applying acquired cognitive coping strategies in real-life contexts.

Pain neuroscience education (PNE), left–right recognition training, and implicit/explicit imagery exercises contributed to the correction of pain prediction errors and the reduction of avoidance behaviors. Furthermore, the observed reduction in PFC hyperactivation via fNIRS may indicate a shift toward more efficient regulation of cortical responses. These findings support the potential effectiveness of a biopsychosocial integrative approach [57,58]. The combined use of repeated sensorimotor stimulation and PNE may have contributed to the attenuation of central sensitization and the enhancement of descending pain-modulatory networks [9,59,60,61].

Therefore, the decreased PFC and DLPFC activity, changes in psychological factors, and improvement of PPT and cervical function observed in this study show that multiple pain control pathways were effectively engaged, suggesting that this approach has the potential, as a clinical intervention strategy, to induce changes in both pain perception and nervous system response.

This case study demonstrates that an integrative intervention may positively impact the functional, psychological, and neurological dimensions of chronic neck pain. Future studies should incorporate long-term follow-up and randomized controlled trials to verify the sustainability and generalizability of these effects. In particular, detailed fNIRS-based analysis under various conditions and the development of personalized intervention strategies tailored to individual psychological and neurophysiological profiles are warranted. Such research would contribute to establishing a more systematic and evidence-based approach to managing chronic pain.

### 5.5. Limitation

This study has inherent limitations due to its single-case design with brief intervention and follow-up periods, which restricts the generalizability and long-term interpretation of the results. The fNIRS outcomes may also be affected by individual and environmental variability. Selection and publication bias are possible. Future research should include larger sample sizes, randomized controlled trials, and long-term, multidimensional assessments to confirm and expand upon these findings.

## 6. Conclusions

This study observed that MT and pain neuroscience-based sensorimotor retraining may have had beneficial effects on headache, pain-related disability, negative psychological factors, functional outcomes, and CNS activation patterns in a patient with CNP. These findings suggest that an integrative approach—addressing physiological, psychological, and neurological mechanisms—may contribute to effective pain management. However, caution is warranted in generalizing these results due to the single-subject design. Further studies are needed to confirm the clinical applicability and long-term outcomes of this intervention.

## Figures and Tables

**Figure 1 healthcare-13-01734-f001:**
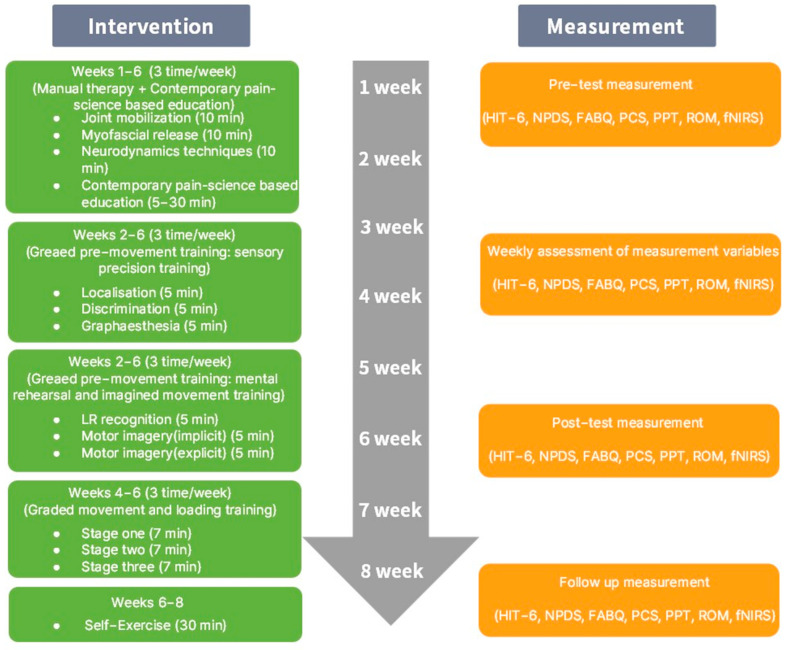
Flow chart of the study design and methods.

**Figure 2 healthcare-13-01734-f002:**
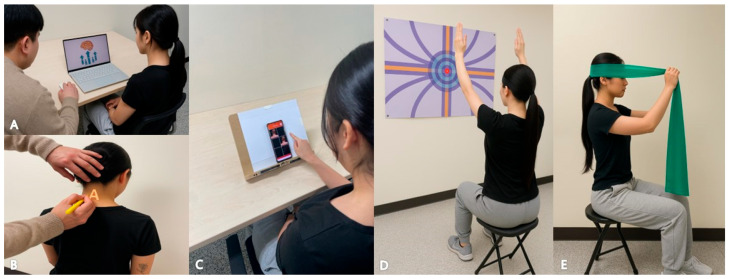
Pain neuroscience-based sensorimotor retraining. (**A**) Pain neuroscience education. (**B**) Graded pre-movement training involving graphesthesia. (**C**) Mental rehearsal through left–right limb recognition. (**D**,**E**) Graded movement and load-based training in Stage 2 of the training protocol.

**Figure 3 healthcare-13-01734-f003:**
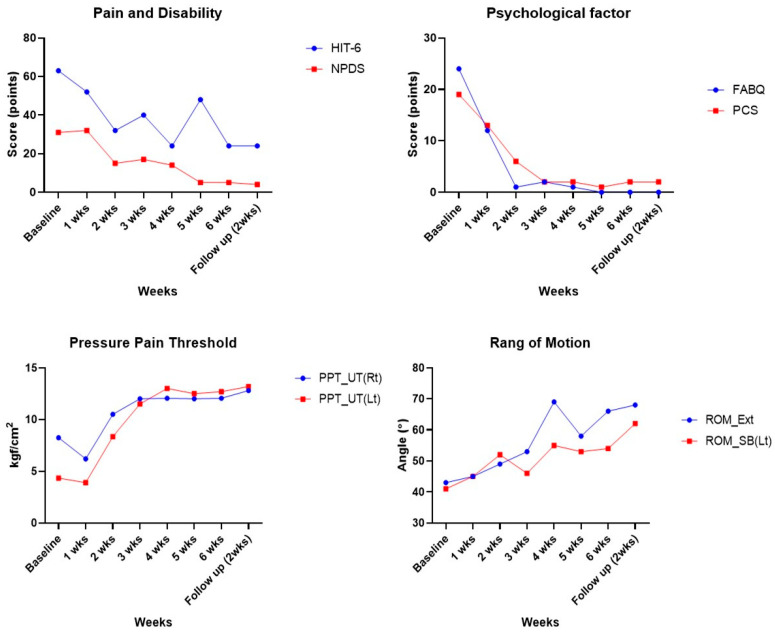
Changes in pain, disability, PPT, psychological factor, and ROM over 6 weeks and during follow-up.

**Figure 4 healthcare-13-01734-f004:**
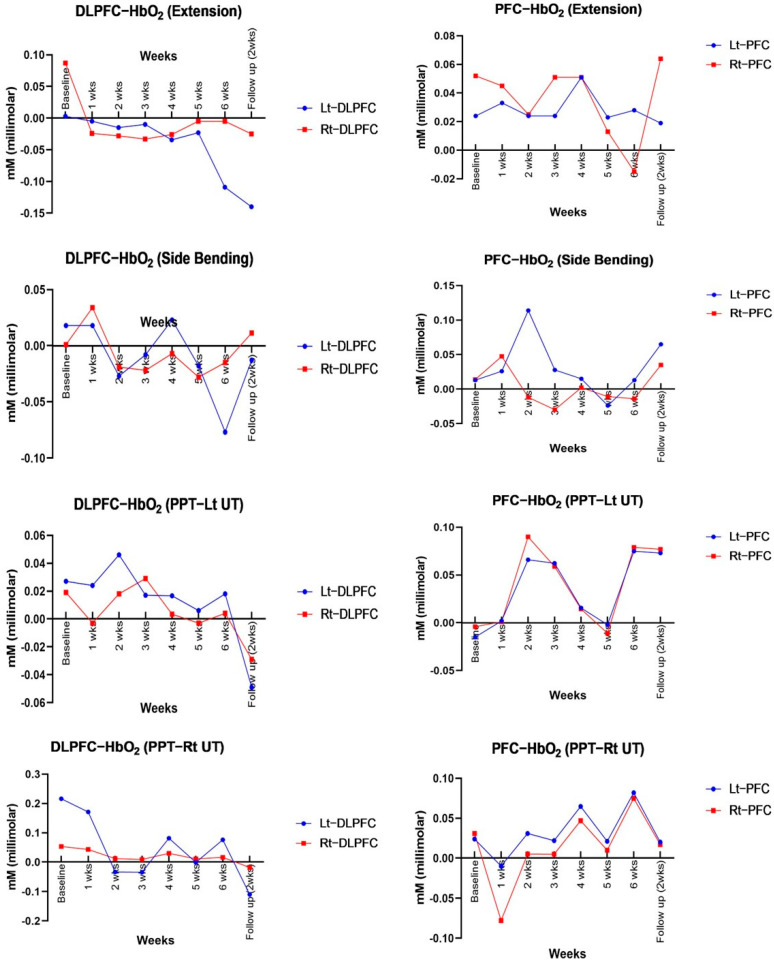
Changes in brain function (fNIRS) over 6 weeks and during follow-up.

**Table 1 healthcare-13-01734-t001:** Clinical and demographic characteristics of the participant.

Variable	Value
Age (years)	25
Height (cm)	158
Weight (kg)	60
HIT-6 (score)	63
NPDS (score)	31
FABQ (score)	24
PCS (score)	19

Note: Headache Impact Test-6 (HIT-6), Neck Pain and Disability Scale (NPDS), Fear-Avoidance Beliefs Questionnaire (FABQ), and Pain Catastrophizing Scale (PCS).

**Table 2 healthcare-13-01734-t002:** Outcomes of pain, disability, and psychological factors during the intervention period.

	Baseline	1 wks	2 wks	3 wks	4 wks	5 wks	6 wks	Follow-Up (2 wks)
HIT-6	63	52	32	40	24	48	24	24
NPDS	31	32	15	17	14	5	5	4
FABQ	24	12	1	2	1	0	0	0
PCS	19	13	6	2	2	1	2	2

Note: HIT-6: Headache Impact Test-6, NPDS: Neck Pain and Disability Scale, FABQ: Fear-Avoidance Beliefs Questionnaire, and PCS: Pain Catastrophizing Scale (Unit: Score).

**Table 3 healthcare-13-01734-t003:** Outcomes of pressure pain threshold and cervical range of motion during the intervention period.

			Baseline	1 wks	2 wks	3 wks	4 wks	5 wks	6 wks	Follow-Up (2 wks)
PPT	UT	Lt	4.34	3.9	8.36	11.5	13	12.5	12.7	13.2
		Rt	8.25	6.2	10.5	12	12.06	12	12.5	12.8
ROM	Ext		43	45	49	53	69	58	66	68
	Left SB		41	45	52	46	55	53	54	62

Note: PPT: Pressure Pain Threshold (Unit: Kgf/cm^2^), ROM: Range of Motion (Unit: Degrees), UT: Upper Trapezius, Lt: Left, Rt: Right, Ext: Extension, and SB: Side Bending.

**Table 4 healthcare-13-01734-t004:** Outcomes of fNIRS measurements during extension and left side-bending tasks over the intervention period.

		Baseline	1 wk	2 wks	3 wks	4 wks	5 wks	6 wks	Follow-Up (2 wks)
Ext	Lt DLPFC	0.003	−0.005	−0.015	−0.010	−0.0343	−0.023	−0.109	−0.140
	Rt DLPFC	0.087	−0.024	−0.028	−0.033	−0.026	−0.005	−0.005	−0.025
	Lt PFC	0.024	0.033	0.024	0.024	0.051	0.023	0.028	0.019
	Rt PFC	0.052	0.045	0.025	0.051	0.051	0.013	−0.015	0.064
Lt SB	Lt DLPFC	0.018	0.018	−0.027	−0.008	0.023	−0.018	−0.077	−0.0128
	Rt DLPFC	0.001	0.034	−0.019	−0.022	−0.007	−0.028	−0.015	0.0113
	Lt PFC	0.013	0.026	0.114	0.0278	0.015	−0.024	0.0219	0.0653
	Rt PFC	0.014	0.0474	−0.012	−0.030	0.002	−0.011	−0.014	0.0359

Note: Ext: Extension, Lt: Left, SB: Side Bending, fNRIS: functional Near-Infrared Spectroscopy, DLPFC: Dorsolateral Prefrontal Cortex, and PFC: Prefrontal Cortex (Unit: mMol).

**Table 5 healthcare-13-01734-t005:** Outcomes of fNIRS measurements during pressure pain threshold tasks over the intervention period.

			Baseline	1 wk	2 wks	3 wks	4 wks	5 wks	6 wks	Follow-Up (2 wks)
PPT	Lt UT	Lt DLPFC	0.027	0.024	0.046	0.017	0.0166	0.006	0.018	−0.049
		Rt DLPFC	0.019	−0.003	0.018	0.029	0.0034	−0.003	0.004	−0.029
		Lt PFC	−0.015	0.002	0.066	0.0623	0.0156	−0.002	0.075	0.073
		Rt PFC	−0.004	0.001	0.090	0.059	0.0145	−0.011	0.079	0.077
	Rt UT	Lt DLPFC	0.216	0.171	−0.033	−0.034	0.0816	−0.001	0.076	−0.110
		Rt DLPFC	0.053	0.043	0.012	0.009	0.0297	0.011	0.016	−0.018
		Lt PFC	0.024	−0.010	0.031	0.022	0.0647	0.021	0.082	0.020
		Rt PFC	0.031	−0.078	0.005	0.0049	0.0469	0.010	0.075	0.017

Note: Lt: Left, Rt: Right, fNRIS: functional Near-Infrared Spectroscopy, UT: Upper Trapezius, DLPFC: Dorsolateral Prefrontal Cortex, and PFC: Prefrontal Cortex (Unit: mMol).

## Data Availability

The data presented in this study are available on request from the corresponding author.

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
