# Peer review of "Multidimensional Effects of Manual Therapy Combined with Pain Neuroscience-Based Sensorimotor Retraining in a Patient with Chronic Neck Pain: A Case Study Using fNIRS"

_healthcare, 2025, doi:10.3390/healthcare13141734_

Round 1
Reviewer 1 Report
Comments and Suggestions for Authors
Respected authors
My comments on your paper
1-Title
The title of the paper should be changed to convey the clear message that this is a case report - That is ,this is a study of a single patient
The overall title of the paper has to be rewritten for more clarity
2- As this is a case report or a study on a single case , are u justified in using material and method section?
3- You have mentioned in abstract in both materials and methods and results that there was a significant improvement in patients overall symptoms ? without mentioning any scales or grading ?
4- The patient had chronic neck pain and radiation to left upper limb ? there is no mention of any consultation with a primary specalist ( orthopaedic or neurology) so is it assumed all the investgastions and evaluation were done? is this correct
5- The discussion and conclusion sections are too extensive and it is very hard for the reader to clearly understand what is the take home message from your study on a single patient .
6 - You have not clearly mentioned in discussion or conclusion , what exactly were the reasons for the patients symptoms and signs, how did u diagnose these and how your intervention methodlogies helped ...... plz explain with scientific reasons with refernces .
Thank you
Author Response
Dear Reviewer,
Thank you for your valuable comments and suggestions.
We have carefully revised the manuscript and prepared a point-by-point response to your feedback.
Please find the attached revision response document for your review.
Best regards,
Ju-hyeon Jung PT, PhD.
Department of Physical Therapy, College of Nursing, Healthcare Sciences and Human Ecology, Dong-Eui University, Busan, Republic of Korea, 47340

Reviewer 2 Report
Comments and Suggestions for Authors
Chronic neck pain is a multifactorial condition characterized by the interplay of pain, psychological stress, and functional impairment interact.
AUTHORS aimed to evaluate the effects of an integrated intervention combining manual therapy and graded sensory motor retraining on pain, psychological factors, functional performance, and cerebral cortical activity in a patient with chronic neck pain.
The patient was a 25-year-old female patient with chronic neck pain and recurrent headache. The intervention comprised manual therapy and pain neuroscience-based sensorimotor retraining, performed three times per week for 6 weeks. The patient’s six-item Headache Impact Test, Neck Pain and Disability Scale, and Pain Catastrophizing Scale scores; pressure-pain threshold (PPT); and cervical range of motion (CROM) were evaluated before and after the intervention, and functional near-infrared spectroscopy (fNIRS) was employed to analyze changes in the activity of the prefrontal cortex and dorsolateral prefrontal cortex (DLPFC) during cervical movement and PPT performance.
AUTHORS results showed that: (1) Post-intervention, pain intensity, headache impact, and negative psychological factors were reduced, and the PPT, CROM, and functional performance ability improved. (2) Functional near-infrared spectroscopy revealed a tendency for decreased oxygenation of the DLPFC during task performance, suggesting alleviation of the cerebral cortex burden associated with pain processing. (3) (4) This effect persisted at the follow-up assessment 2 weeks post-intervention.
AUTHORS concluded that their findings suggest that a 6-week integrated intervention targeting biopsychosocial pain mechanisms may be feasible for pain reduction, functional improvement, and enhancement of neurological modulation mechanisms in patients with chronic neck pain.
The authors address a very important and impactful topic
These are my suggestions in a purely academic spirit:
1) The introduction for a case study is adequate. I would only like it if some passages were more detailed, developing the contribution of the reference. See for example “and biological mechanisms of pain, enabling reconstruction of their pain experience [8,9].”
2) Table one presents heterogeneous data, such as age and the score of the quationary and from an editorial point of view it does not hold much.
3) Figure 1 is very interesting but it should be illustrated in the text in a broader way
4) Wide discussion. The only suggestion is to concentrate in one paragraph (the info is scattered) the trajectory of future research that the case study suggests.
Overall the case study was conducted in a broad and detailed way with all the sections developed in a broad and coherent flow of rationale. I hope my suggestions are useful for final refinements.
I suggest continuing on this promising path
Author Response

(The authors gave the same response as above.)

Reviewer 3 Report
Comments and Suggestions for Authors
Dear Editor,
Thank you for your kind invitation to review this manuscript.
In this case report, the authors aim to present the treatment outcomes of a young patient with chronic neck pain.
My comments regarding the manuscript are listed below:
-
The keywords should be selected from MeSH terms.
-
The introduction section can be slightly shortened.
-
The use of references throughout the manuscript should be carefully checked. Some older references can be replaced with more up-to-date ones.
-
Unnecessary and repetitive sentences within the manuscript should be removed.
For example:
“Table 1 presents the general characteristics of the participant and the Chronic Pain-related Assessment Scale score.” -
The methodology is explained in detail, and objective evaluation results have been used.
-
In the intervention section, I recommend specifying within the manuscript which healthcare professional administered the treatment.
-
In Table 3, providing all abbreviations below the table will make it easier for readers to understand.
-
In the Results section, do not repeat the data already presented in tables and figures in the main text.
Please remove repetitive expressions and revise the text accordingly. -
In the Discussion section, do not restate your own results; I recommend revising this section.
Discuss the changes you observed in relation to the underlying neurophysiological mechanisms.
Author Response

(The authors gave the same response as above.)

Reviewer 4 Report
Comments and Suggestions for Authors
1. The introduction repeats similar explanations of central sensitization in multiple places using slightly varied phrasing. This repetition adds to the word count without contributing new information.
Could condensing the background discussion on central sensitization improve clarity and efficiency?
2. Several statements regarding the limited long-term effects of manual therapy lack specific citations or are not clearly linked to referenced sources. This may weaken the argument for the novel intervention being proposed.
Wouldn't strengthening the reference support for these claims improve the credibility of the rationale?
3. The methodology for fNIRS use is explained but lacks important technical details such as calibration procedures. Could more information be added here?
4. table 4 is overwhelming visually. This makes it difficult for the reader to extract meaningful trends or interpret changes across conditions.
Separating this table into smaller, categorized tables or adding graphical elements make the data easier to interpret.
5. The results section would benefit from clearer synthesis of fNIRS data—at present, the numerical outputs dominate, but higher-level summarization or trend interpretation is lacking. This could leave readers unclear on the significance of the results. Please consider adding a better summary to enhance interpretability.
6. Statements such as “this means that…” or “this shows that…” in the discussion are occasionally too definitive given the single-subject design and inherent variability of fNIRS. This may risk overstating the strength of the findings.
Could replacing absolute phrasing with more tentative language (e.g., “suggests,” “may indicate”) improve interpretive balance?
7. The study is presented as a proof of concept, but there are occasional moments—especially in the conclusion—where generalization is implied. Given the single-subject design and short follow-up, such statements should be more cautiously framed. Would reinforcing the exploratory nature of the findings help clarify the study’s scope and avoid overgeneralization?
8. There are a few small grammar slips here and there—like missing or extra articles (“a,” “the”). A quick language edit or light copyedit help smooth things out and boost the manuscript’s clarity?
Author Response

(The authors gave the same response as above.)

Round 2
Reviewer 1 Report
Comments and Suggestions for Authors
Respected authors
The changes as suggested have been made .
Thank you
Reviewer 4 Report
Comments and Suggestions for Authors
Thank you for accepting the feedback and implementing it. The paper has improved a lot and much readable.